# Correlational Study of Aminopeptidase Activities between Left or Right Frontal Cortex versus the Hypothalamus, Pituitary, Adrenal Axis of Spontaneously Hypertensive Rats Treated with Hypotensive or Hypertensive Agents

**DOI:** 10.3390/ijms242116007

**Published:** 2023-11-06

**Authors:** Isabel Prieto, Ana Belén Segarra, Inmaculada Banegas, Magdalena Martínez-Cañamero, Raquel Durán, Francisco Vives, Germán Domínguez-Vías, Manuel Ramírez-Sánchez

**Affiliations:** 1Department of Health Sciences, University of Jaén, 23071 Jaén, Spain; iprieto@ujaen.es (I.P.); asegarra@ujaen.es (A.B.S.); ibanegas@ujaen.es (I.B.); canamero@ujaen.es (M.M.-C.); 2Department of Physiology, Faculty of Medicine, University of Granada, 18071 Granada, Spain; rduran@ugr.es (R.D.); fvives@ugr.es (F.V.); 3Department of Physiology, Faculty of Health Sciences, Ceuta Campus, University of Granada, 18071 Granada, Spain; germandv@ugr.es

**Keywords:** brain asymmetry, neuro-endocrine asymmetry, neuro-visceral integration, hypertension, renin–angiotensin system, aminopeptidases, frontal cortex, hypothalamus, pituitary, adrenal

## Abstract

It has been suggested that the neuro-visceral integration works asymmetrically and that this asymmetry is dynamic and modifiable by physio-pathological influences. Aminopeptidases of the renin–angiotensin system (angiotensinases) have been shown to be modifiable under such conditions. This article analyzes the interactions of these angiotensinases between the left or right frontal cortex (FC) and the same enzymes in the hypothalamus (HT), pituitary (PT), adrenal (AD) axis (HPA) in control spontaneously hypertensive rats (SHR), in SHR treated with a hypotensive agent in the form of captopril (an angiotensin-converting enzyme inhibitor), and in SHR treated with a hypertensive agent in the form of the L-Arginine hypertensive analogue L-NG-Nitroarginine Methyl Ester (L-NAME). In the control SHR, there were significant negative correlations between the right FC with HPA and positive correlations between the left FC and HPA. In the captopril group, the predominance of negative correlations between the right FC and HPA and positive correlations between the HPA and left FC was maintained. In the L-NAME group, a radical change in all types of interactions was observed; particularly, there was an inversion in the predominance of negative correlations between the HPA and left FC. These results indicated a better balance of neuro-visceral interactions after captopril treatment and an increase in these interactions in the hypertensive animals, especially in those treated with L-NAME.

## 1. Introduction

The renin–angiotensin system (RAS) is essential in regulating blood pressure. The analysis of its enzymatic constituents, particularly the aminopeptidases glutamyl aminopeptidase (GluAP), alanyl aminopeptidase (AlaAP), and cystinyl aminopeptidase (CysAP) (generically referred to herein as angiotensinases), offers us a dynamic insight into the metabolism of the important peptides of the system, such as the angiotensins (Ang) Ang II, Ang III, and Ang IV. The RAS is activated in genetically hypertensive rats (SHR) [1,2]. The organisms work within a neuro-visceral integration [3], and recently, it has been suggested that this integration, mediated probably through the autonomic nervous system, works asymmetrically and that this asymmetry can be modified depending on variations under physiological or pathological conditions [4].

### 1.1. Role of Aminopeptidases in the Renin–Angiotensin System

The RAS was traditionally thought to be only involved in the regulation of blood pressure, but nowadays, it is related to multiple other functions, including the performance of its peptides as neurotransmitters/neuromodulators [5]. Briefly, focusing on the role of aminopeptidases in this system (Figure 1), the angiotensin-converting enzyme (ACE) acts after the action of renin (an endopeptidase that hydrolyzes the angiotensinogen, a precursor protein of the system), producing angiotensin I (Ang I). ACE is a peptidyl dipeptidase that cleaves the carboxy-terminal dipeptide (His-Leu) to give rise to Ang II. Aminopeptidase A (glutamate aminopeptidase) acts on this peptide by cleaving the amino-terminal Asp to give rise to Ang III, and on this, via the action of aminopeptidase M (AlaAP) or aminopeptidase B (ArgAP), Ang IV is generated. 

Each of the mentioned angiotensins, produced via the action of the various peptidases, performs different functions after binding to various receptors. Thus, after the binding of Ang II to the AT_1_ receptor, the actions that have long been considered to be the main ones of the system are carried out, such as vasoconstriction, blood pressure regulation, or the progressive deterioration of organs such as the kidneys or heart. Ang III can bind to the AT_1_ or AT_2_ receptor, and in both cases, the action is similar but less potent than that of Ang II. Notably, Ang III is the most abundant and active form of Ang in the brain [6]. Ang IV, after binding to the AT_4_ receptor, is an important local regulator of blood flow; it improves glucose tolerance and has been reported to improve cognitive functions. It is worth noting that this receptor protein has been identified as insulin-regulated aminopeptidase (IRAP) (cystinyl aminopeptidase, oxytokinase or vasopressinase), which is widely distributed in tissues and plays a role in the control of blood pressure and water balance [7], or cognitive functions [8]. Finally, antihypertensive effects have been described for des-Asp-Ang I (Ang 2–10), which counteracts the action of Ang II and Ang III and acts at the renal [9] and brain [10,11] levels. Therefore, the functional regulation of such angiotensins, both centrally and peripherally, lies essentially in the action of aminopeptidases, which we refer to herein as angiotensinases, although they can also act on peptides other than angiotensins, such as, among others, enkephalins [12].

### 1.2. Objectives

We selected the study of aminopeptidase activities because they are the most abundant proteolytic enzymes in the nervous system [13] and due to their key role in the inactivation and generation of important RAS peptides (Figure 1). Partial data have been previously described regarding the behavior of angiotensinases under treatments with captopril or the L-Arginine hypertensive analogue L-NG-Nitroarginine Methyl Ester (L-NAME) in the left FC and right FC [14], as well as regarding the behavior in the hypothalamus [15] and their behavior in the PT and AD [16]. We currently have the correlational data of such activities, which allowed us to assess the bilateral behavior of the interaction between angiotensinase activities in an axis integrated by the left or right frontal cortices with the tissues of the hypothalamus–pituitary–adrenal axis (HPA) in genetically hypertensive rats (SHR) (control), in SHR rats after the administration of a hypotensive treatment with the ACE inhibitor captopril, and in SHR rats after the administration of a hypertensive treatment with L-NAME, and the results of this study may offer us a greater understanding about the trends in the interactions between these tissues with such opposite treatments. The increase in blood pressure in L-NAME-treated animals was explained by the inhibition of nitric oxide synthase and by the activation of the sympathetic nervous system [17].

## 2. Results

The animals used in the present investigation showed the following systolic blood pressure levels with the various treatments: control SHR—206.7 ± 6.8 mmHg, captopril—159.3 ± 5.2 mmHg, L-NAME—234.7 ± 4.1 [14]. In the control SHR, the results denoted significant negative correlations between the right FC and the HT (right FC Sol AlaAP vs. hypothalamus Sol AlaAP: r = −0.74, *p* = 0.03), PT (right Sol AlaAP vs. pituitary Memb AlaAP: r = −0.74, *p* = 0.03), and AD (right Sol CysAP vs. adrenal Memb CysAP: r = −0.80, *p* = 0.01) and positive correlations between the left FC and tissues of the axis (left FC Sol GluAP vs. hypothalamus Memb AlaAP: r = +0.77, *p* = 0.02; left FC Memb vs. pituitary Sol GluAP: r = +0.89, *p* = 0.003; left FC Sol CysAP vs. adrenal Sol AlaAP: r = +0.75, *p* = 0.03), as well as a diversity of positive and negative correlations between the HT, PT, and AD. Regarding intra-tissue correlations, two negative ones were observed in adrenals. The rest of the intra-tissue correlations were positive (Appendix A) (Figure 2) (Sol, soluble; Memb, membrane-bound). 

In the captopril-treated animals, the predominance of negative correlations between the right FC and the different tissues of the axis (right FC Memb GluAP vs. hypothalamus Sol GluAP: r = +0.70, *p* = 0.05; right FC Memb CysAP vs. pituitary Memb GluAP: r = −0.72, *p* = 0.02; right FC Memb GluAP vs. adrenal Memb GluAP: r = −0.82, *p* = 0.01) and positive correlations between these and the left FC were maintained (left FC Memb CysAP vs. hypothalamus Sol GluAP: r = +0.89, *p* = 0.003; left FC Memb AlaAP vs. pituitary Sol AlaAP: r = +0.70, *p* = 0.05; left FC Memb AlaAP vs. adrenal Sol GluAP: r = +0.72, *p* = 0.02). Furthermore, in this group, there were inter-hemispheric correlations between the right and left FC (left FC Memb GluAP vs. right FC Sol GluAP: r = −0.70, *p* = 0.05; left FC Memb GluAP vs. right FC Memb CysAP: r = +0.78, *p* = 0.02 [14]), and there was a clear reduction in interactions between the HT, PT, and AD, all of them positive. One negative intra-hypothalamic significant correlation was observed, while the rest of the intra-tissue correlations were positive. No intra-pituitary correlations were observed in this group (Appendix A) (Figure 3).

In the animals treated with L-NAME, a radical change in all types of interactions was observed. In general, there was an increase in all of them, and particularly, there was a change to a predominance of negative correlations between the left FC and the HT, PT, and AD. No significant correlations were observed between the right FC and hypothalamus, but there were correlations with pituitary (right FC Memb CysAP vs. pituitary Sol CysAP: r = +0.73, *p* = 0.03; right FC Memb AlaAP vs. pituitary Memb AlaAP: r = −0.83, *p* = 0.01) and adrenal (right FC Memb CysAP vs. adrenal Sol AlaAP: r = +0.70, *p* = 0.05) between the left FC and hypothalamus (left FC Sol AlaAP vs. hypothalamus Sol CysAP: r = +0.74, *p* = 0.03; left FC Sol AlaAP vs. hypothalamus Sol GluAP: r = +0.78, *p* = 0.02), left FC and pituitary (left FC Sol AlaAP vs. pituitary Sol AlaAP: r = −0.78, *p* = 0.02; left FC Memb AlaAP vs. pituitary Sol CysAP: r = −0.77, *p* = 0.02; left FC Memb GluAP vs. pituitary Memb GluAP: r = −0.80, *p* = 0.01), and left FC versus adrenal (left FC Memb CysAP vs. adrenal Sol CysAP: r = +0.83, *p* = 0.01; left FC Sol CysAP vs. adrenal Memb AlaAP: r = −0.71, *p* = 0.04; left FC Sol GluAP vs. adrenal Memb AlaAP: r = −0.74, *p* = 0.03). Similarly, a marked increase in the correlations of the HT, PT, and AD with each other was observed. Only significant positive intra-tissue correlations were observed—particularly abundant in the right FC and hypothalamus (Appendix A) (Figure 4).

## 3. Discussion

As indicated previously, the activity of the enzymes studied is not limited to angiotensins; however, considering the groups of animals analyzed, we limited the scope of our discussion to their role in the RAS. The main focus of the present investigation lies in the marked difference observed between the SHR control animals and the hypotensive treatment with captopril or the hypertensive one with L-NAME in the positive or negative character of the predominance of the left or right FC correlation and in the level of correlation of the left or right FC with the rest of the tissues, as well as in the positive or negative character and in the level of inter-tissue correlation between the HT, PT, and AD.

A greater understanding of the results can be obtained from Figure 5. First, in the control (SHR) group, we found that only significant negative correlations between the right FC and the rest of the tissues were observed; only significant positive correlations between the left FC and the rest of the tissues were observed; a majority of significant positive inter-tissue correlations were observed between the HT, PT, and AD; and the majority of significant intra-tissue correlations were positive. Second, in the captopril group, in general, the observed number of significant correlations were reduced; a majority of significant negative inter-tissue correlations with right FC and positive with left FC were maintained; some slight significant inter-hemispheric correlations appeared; there was a notorious decrease in the number of significant inter-tissue correlations between the HT, PT, and AD; and the majority of significant intra-tissue correlations were positive. Finally, in the L-NAME group, a generalized disruption of the positive or negative character and the number of significant correlations was observed in comparison with the other two groups; there was a general increase in the number of significant inter-tissue and intra-tissue correlations; and the highest number of significant negative correlations between the FCs and the rest of tissues changed to be in the left FC in comparison with the other two groups. All the significant intra-tissue correlations were positive, especially in the right FC and HT, where the number and level of correlations increased.

Brain asymmetry offers us a complex picture. It is not a static concept but a dynamic one that is modifiable depending on the location or the function involved, the variations in the external environmental, and internal physio-pathological conditions. In addition, all of the above can be extended to the concept of neuro-visceral integration, which is also asymmetric and modifiable [4]. Considering the results locally in the left and right frontal cortices, in the normotensive WKY animals, captopril induced greater inter-hemispheric correlations, especially those of a positive nature, compared to the control animals or those treated with L-NAME. In the SHR, inter-hemispheric correlations also appeared in the animals treated with captopril—correlations that were not observed in the SHR controls or those treated with L-NAME [14]. In [14], the bilateral behavior of aminopeptidase activities in the frontal cortices was also analyzed in normotensive animals, and the results of this study, as was the case in our study involving SHR, showed that the CAP group had the highest number of inter-hemispheric correlations.

In addition to the above, if we consider the correlations between the left or right FC and the rest of the tissues, in the animals treated with captopril, the negative correlation between the right FC and hypothalamus that existed in the SHR controls is lost and the correlations between HT, PT, and AD decrease drastically. The greatest change was observed after treatment with L-NAME; the correlations of the left and right frontal cortices with the rest are inverted with respect to the other two groups, as a large number of negative correlations now appear between the left FC, PT, and AD and, again, as in the control SHR, a large number of correlations are observed between the HT, PT, and AD. Some of these partial results between the PT and AD have already been observed previously [16]. All of the above could suggest that the significant changes in correlations observed with respect to the other two groups with pathological blood pressure levels could also be involved in the beneficial effect of captopril, especially if we consider the animals treated with L-NAME, in which blood pressure reached such high levels to the point where it compromised the lives of the animals. All this could denote the organism’s attempt to restore neuro-visceral/viscero-neural communication to try to recover the altered integration in conditions of hypertension. There is no doubt that these important changes involving the frontal cortex may lead to cognitive modifications in parallel to integral alterations in the functioning of the organism [4]. Other interesting results remark the complexity of the asymmetry in neuro-visceral integration. Thus, in the control group of hypertensive animals, a similar number of positive and negative correlations between the right FC and plasma was observed. However, after treatment with captopril, there was a significant increase in the number of negative correlations but now with the left FC [18]. Also, when analyzing the interaction between the left and right FC with the left ventricle, a similar number of positive and negative correlations with the left FC of the control SHR was observed, but after treatment with captopril, the number of correlations, especially those of a positive character, with the right FC increased [19].

All of the above connects the cognitive functions that take place between the left and right FC with the rest of the tissues, and vice versa; the functional states of these tissues connect with the functions carried out by the FC, all of which are largely connected bidirectionally by the autonomic nervous system or through neuro-endocrine connections [4]. The frontal lobe is involved in numerous cognitive functions [20,21] that, overall, are processed asymmetrically [4,22]. The left and right frontal cortex exhibit great diversity in their connections with peripheral tissues [4]. The fact that brain asymmetry has become a broader concept of “asymmetric neuro-visceral integration” requires a multidisciplinary analysis for a better understanding [4,23]. Potential limitations may underlie studies on brain asymmetry. For example, analyzing the possibility that asymmetry could be explained in part by interhemispheric differences in axonal conduction, Partadiredja et al. [24] found no interhemispheric differences in the number, caliber, or type of axons. Also, focusing on the diversity of functions in which the enzymes analyzed in the present work may be involved, Wu et al. [25] demonstrated that the Sol and Memb activity of AlaAP and CysAP present an asymmetric behavior in the hippocampus of rats that is associated with paw preference. Interestingly, proteomic studies reveal asymmetries in the hippocampus. For example, a greater abundance of enzymes related to cellular energy metabolism in the right hippocampus than in the left and, in contrast, a higher concentration of proteins was observed in astrocytes from the left hippocampus rather than from the right [26]. Furthermore, in relation to the anterior reference, we must take into account that SHR exhibit vascular brain disorders and neuroinflammation [27] and that SolAlaAP (which converts Ang III to Ang IV) (Figure 1) increases significantly in astrocytes, increasing Ang IV and exacerbating neuroinflammation [28]. Therefore, our results for the group of animals treated with L-NAME could be related to exacerbated neuroinflammation produced by the hypertensive treatment.

## 4. Materials and Methods

### 4.1. Animals, Treatments, Blood Pressure Determination, and Ethical Approval

Twenty-four spontaneously hypertensive rats (SHR) (Charles River Laboratories, Barcelona, Spain) with a weight range between 100 and 150 g when the experiments began were used for this study. The animals were randomly divided into three groups, each with eight animals: control SHR, captopril-treated SHR, and L-NAME-treated SHR. Captopril (Sigma-Aldrich, St Louis, MO, USA; 100 mg/kg/day) and L-NAME (Sigma-Aldrich St Louis, MO, USA; 70 mg/kg/day) were added to drinking water for twenty-eight days. The doses and timing of administration have been previously described [29,30]. In order to avoid modifications due to circadian and seasonal rhythms, all the experiments were carried out between the months of April and July between 9:00 a.m. and 12:00 p.m [31,32]. Using previously described methods [16,33], systolic blood pressure was determined by carrying out tail-cuff plethysmography on unanesthetized rats (LE 5001-Pressure Meter; Letica SA, Barcelona, Spain) maintained in plastic holders at 37 °C. The experiments were carried out according to the European Communities Council Directive 86/609/EEC, and our study protocol was approved by the bioethics committee of the University of Jaén.

### 4.2. Surgical Procedure and Obtaining Tissue Samples

Once the treatment period ended and the blood pressure levels were obtained, the animals were fully perfused with saline under equithensin anesthesia (2 mL/kg body weight) [42.5 g/L chloral hydrate dissolved in 19.76 mL ethanol, 9.72 g/L Nembutal (Sigma-Aldrich) 0.396 L/L propylenglycol and 21.3 g/L magnesium sulfate in distilled water]. The left and right frontal cortices and the hypothalamus were obtained according to the stereotaxic coordinates obtained from the Paxinos and Watson atlas [34]. The frontal cortex (separating left and right) was dissected from the anterior borders of the frontal lobes to 13.2 mm anterior to the interaural line. The hypothalamus (pooled left and right) was obtained between 7.7 mm and 3.7 mm, anterior to the interaural line. The pituitary (pooled anterior and posterior) and adrenal glands (pooled left and right) were quickly removed (less than 60 s).

### 4.3. Methods for the Determination of Proteins and Enzymatic Activities

In order to obtain the soluble fraction, samples of the different tissues were homogenized in an hypoosmolar medium (10 mmol/l HCl-Tris buffer, pH 7.4) and ultracentrifuged at 100,000× *g* for 30 min at 4 °C. Then, the supernatants were used for soluble proteins and enzyme assays. In order to obtain the particulate fraction, the pellets were re-homogenized in a HCl–Tris buffer (pH 7.4) and 1% Triton X-100 to solubilize membrane proteins. After centrifugation (100,000× *g*, 30 min, 4 °C), the protein levels and activities of the membrane-bound enzymes were measured in the supernatants. To ensure the complete recovery of activity, the detergent was removed from the medium by adding adsorbent polymeric Bio-Beads SM-2 (Sigma) (100 mg/mL) and shaking the samples for 2 h at 4 °C. The activity of the soluble and membrane-bound aminopeptidases, measured as glutamyl- (GluAP), alanyl- (AlaAP), and cystinyl-aminopeptidase (CysAP), was determined fluorometrically using the arylamide derivatives glutamyl-, alanyl-, and cystinyl-β-naphthylamide as substrates as previously described [14,15,16,35,36]. Briefly, GluAP was determined using Glu-β-naphthylamide as a substrate; 10 mL of each supernatant was incubated for 120 min at 37 °C with 1 mL of the substrate solution (2.72 mg/100 mL Glu-β-naphthylamide, 10 mg/100 mL bovine serum albumin (BSA), 10 mg/100 mL dithiothreitol (DTT), and 0.555 g/100 mL CaCl_2_ in 50 mmol/L HCl-Tris, pH 7.4). AlaAP and CysAP were measured using Ala or Cys-β-naphthylamide as substrates so that 10 mL of each supernatant and plasma were incubated for 30 min at 25 °C with 1 mL of the substrate solution, that is, 2.14 mg/100 mL of Ala-β-naphthylamide or 5.53 mg/100 mL of Cys-β-naphthylamide, 10 mg/100 mL BSA and 10 mg/100 mL DTT in 50 mmol/L of phosphate buffer (pH 7.4 for AlaAP), and 50 mmol/L HCl-Tris buffer (pH 6 for CysAP). The reactions were terminated via the addition of 1 mL of 0.1 mol/L of acetate buffer (pH 4.2). The amount of β naphthylamine released as a result of the enzymatic activity was measured fluorometrically at a 412 nm emission wavelength with an excitation wavelength of 345 nm. Proteins were quantified in triplicate using the method of Bradford [37] with BSA as a standard. Specific soluble and membrane-bound aminopeptidase activities were expressed as pmol of the corresponding substrate hydrolyzed per minute per milligram of protein. Fluorogenic assays were linear with respect to time of hydrolysis and protein content [14,16].

### 4.4. Statistical Analysis

In order to analyze intra-tissue and inter-tissue correlations between angiotensinase activities in each group (control, captopril, L-NAME) of the studied SHR, Pearson’s correlation coefficient was computed. Computations were performed using SPSS 13.0 (Chicago, IL, USA) and STATA 9.0 (STATA Corp, College Station, TX, USA). *p* values below 0.05 were considered significant.

## 5. Conclusions

In the group of SHR treated with captopril, the reduction in systolic blood pressure levels was also accompanied by a reduction in the frequency of correlations in comparison with the control group of hypertensive animals and with the group in which an additional increase in systolic blood pressure was induced by treatment with L-NAME. In the latter group, not only there was an increase in all kinds of the observed correlations, but also a general disruption, including a reversal of the asymmetry. The benefits of treating hypertension using captopril may therefore convey a modification in the bilateral neuro-endocrine behavior of neuro-visceral integration.

We have not yet been able to discover a pattern or patterns of bilateral response to patho-physiological changes and how these changes would help restore the lost body homeostasis. Therefore, based on all of the above, the biological meaning of brain asymmetry and asymmetry in neuro-visceral integration is a complex issue that still needs to be explored via multiple research approaches. 

## Figures and Tables

**Figure 1 ijms-24-16007-f001:**
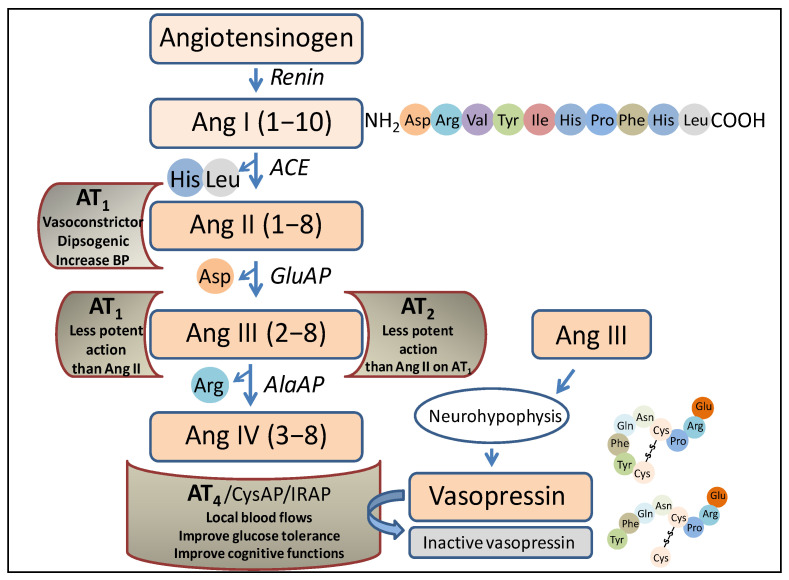
Representation of a part of the renin–angiotensin system in which the enzymes and peptides involved in the present investigation appear, as well as the possible actions of those peptides after binding to their corresponding receptors. Ang, angiotensin; ACE, angiotensin converting enzyme; GluAP, glutamyl aminopeptidase; AlaAP, alanyl aminopeptidase; CysAP, cystinyl aminopeptidase; AT_1_, AT_1_ receptor; AT_2_, AT_2_ receptor; AT_4_, AT_4_ receptor; IRAP, insulin-regulated aminopeptidase, BP, blood pressure.

**Figure 2 ijms-24-16007-f002:**
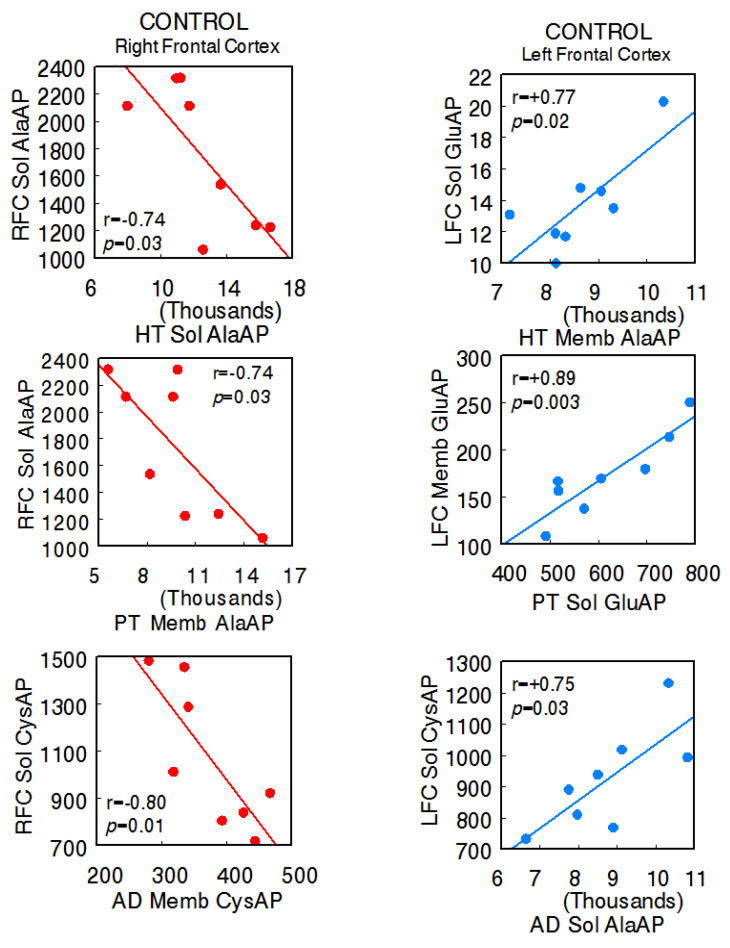
Significant inter-tissue positive or negative correlations obtained between specific soluble (Sol) or membrane-bound (Memb) alanyl-(AlaAP), cystinyl-(CysAP), or glutamyl-(GluAP) aminopeptidase activities, expressed as pmol/min/mg prot, of the left or right frontal cortex versus the same enzymatic activities obtained in the hypothalamus (HT), pituitary (PT), or adrenals (AD) in the control spontaneously hypertensive rats (SHR) (*n* = 8). Negative values are shown in red, while positive values are shown in blue. Pearson’s correlation coefficients (r) and *p* values are indicated in each figure.

**Figure 3 ijms-24-16007-f003:**
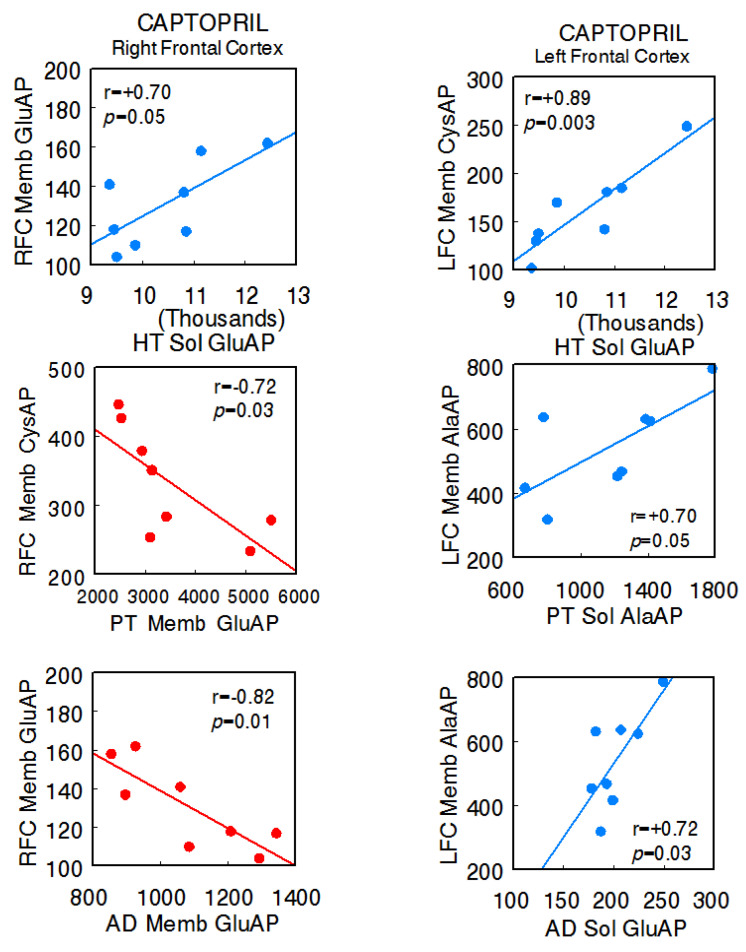
Significant inter-tissue positive or negative correlations obtained between specific soluble (Sol) or membrane-bound (Memb) alanyl-(AlaAP), cystinyl-(CysAP), or glutamyl-(GluAP) aminopeptidase activities, expressed as pmol/min/mg prot, of the left or right frontal cortex versus the same enzymatic activities obtained in the hypothalamus (HT), pituitary (PT), or adrenals (AD) in the captopril-treated spontaneously hypertensive rats (SHR) (*n* = 8). Negative values are shown in red, while positive values are shown in blue. Pearson’s correlation coefficients (r) and *p* values are indicated in each figure.

**Figure 4 ijms-24-16007-f004:**
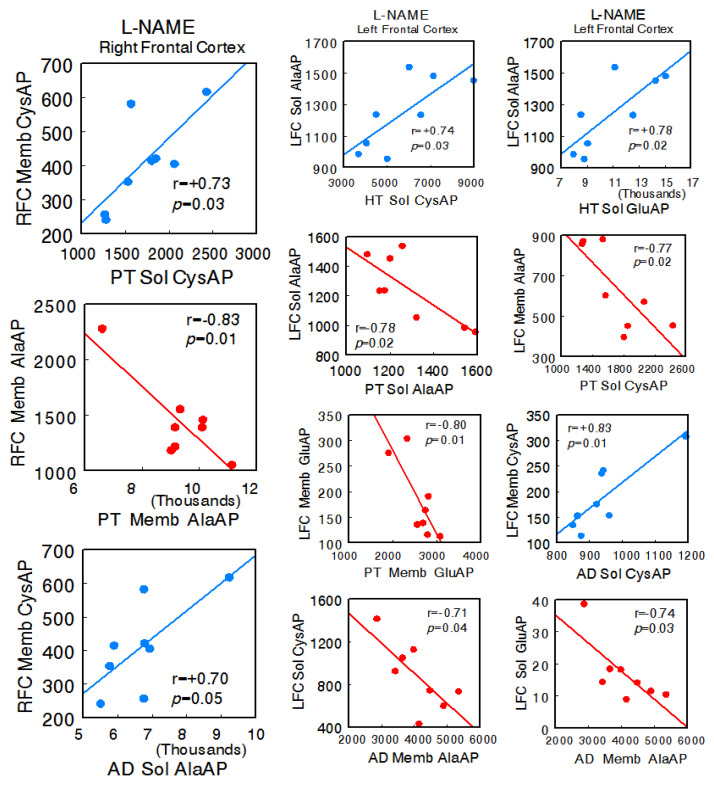
Significant inter-tissue positive or negative correlations obtained between specific soluble (Sol) or membrane-bound (Memb) alanyl-(AlaAP), cystinyl-(CysAP), or glutamyl-(GluAP) aminopeptidase activities, expressed as pmol/min/mg prot, of the left or right frontal cortex versus the same enzymatic activities obtained in hypothalamus (HT), pituitary (PT), or adrenals (AD) in the L-NG-Nitroarginine Methyl Ester (L-NAME)-treated spontaneously hypertensive rats (SHR) (*n* = 8). Negative values are shown in red, while positive values are shown in blue. Pearson’s correlation coefficients (r) and *p* values are indicated in each figure.

**Figure 5 ijms-24-16007-f005:**
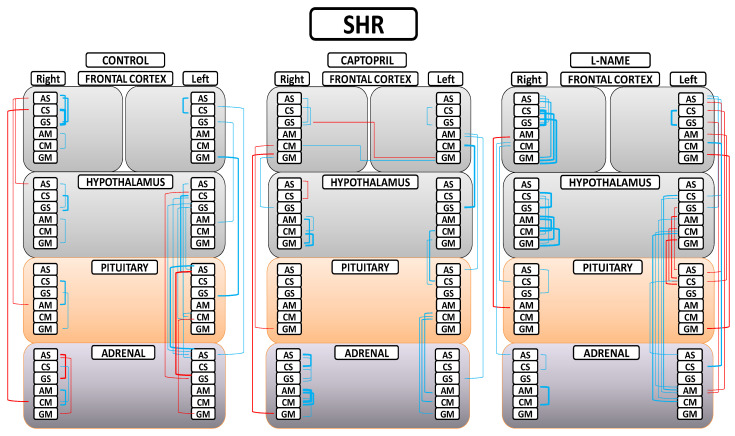
Significant intra- and inter-tissue correlations between angiotensinase activities of right and left frontal cortices, hypothalamus, pituitary, and adrenal gland in the control spontaneously hypertensive rats (SHR), the SHR treated with captopril, and the SHR treated with L-NG-Nitroarginine Methyl Ester (L-NAME). Blue lines—positive significant correlations; red lines—significant negative correlations; line thickness—degree of significance (this figure corresponds to the values indicated in Appendix A). Soluble (AS) or membrane-bound alanyl aminopeptidase (AM), soluble (CS) or membrane-bound cystinyl aminopeptidase (CM) and soluble (GS) or membrane bound glutamyl aminopeptidase (GM). (Appendix A).

## Data Availability

The data presented in this study are available from the corresponding author upon request.

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
