# Peer review of "Correlational Study of Aminopeptidase Activities between Left or Right Frontal Cortex versus the Hypothalamus, Pituitary, Adrenal Axis of Spontaneously Hypertensive Rats Treated with Hypotensive or Hypertensive Agents"

_ijms, 2023, doi:10.3390/ijms242116007_

Round 1

Reviewer 1 Report

Comments and Suggestions for Authors

Dear Editors

The paper needs an extensive revision throughout the text. Though the analysis is extensive, new data calculations and presentations are also suggested. An extensive English spell check revision is required.

I have tried to address all issues in my comments to consider.

Comments on the Quality of English Language

Review

Authors in their manuscript investigate angiotensinase activities in brain cortex and HPA tissues in SHR rats treated with captopril and L-NAME. They found significant changes in correlation between brain FC sides and HPA tissues as indicated of Fig. 4. They have made an extensive correlation analysis form their data.

They conclude that the benefits of treatment of hypertension with captopril may therefore convey a modification in the bilateral neu[1]ro-endocrine behavior of neuro-visceral integration.

Since the study is interesting, the manuscript is not well written, not focused, the presentation of data is not clear and not enough in the present form. The sections have to be focused and cleared. The paper is full of uncertain expressions, e.g. „modified under such conditions”, „In general, there is an increase in all of them”, „We currently have enough data that allow us to assess”, „The rest were positive”, etc, which are better for a personal blog, than for a scientific article.

There are some major issues of the manuscript:

1.     1. The manuscript does not apply for some formal requirements of IJMS: The abstract is too lengthy (200 words are allowed), it is almost double of that. The Conclusions is after Methods. The template does not contain line numbers on the pages, so it is more difficult for the reviever to interpret suggested corrections. The Supplementary material (Table legends) should appear in the Supplement, not in the main text. Etc.

2.    2.  The Introduction looks like a book chapter for university students. It is not focused at all. Please indicate here the up-to-date scientific information about renin-angiotensin system and peptidases studied and their roles also with recent publications. Please omit figures from here and those detailed descriptions of the systems with more well known information.

3.     3. A major issue of the study that what do readers see from data of pure correlation presentation, since there are original data presented. From pure correlations we do not know the sites of changes. It is suggested to show also original data from each tissue and show the main significant data as correlation curves of orig. data (as it is presented from previous publication of Authors in ref. 16).

4.    4.  It is also the question why Authors show an investigation only among hypertensive animals. It would be useful to add a normotensive group for accurate comparisons. It is also required to show blood pressure data of the animal groups in Results and to discuss these data compared to previous studies. Please explain if systolic or mean values are indicated in the test and why these values are measured so high (above 200 mmHg, a data distribution in all groups would be reguired).

5.     5. As it is mentioned on page 8 „Some of these partial results between PT and AD have already been observed previously [16].”, it needs some clarification if Authors used data from previous publications or if they have not published the present data before. In a prev. case maybe it needs some permission requirements.

6.     6. The references are full of older papers. Please give also more recent publications of the area and add to them.

7.    7.  It is also a question how angiotensin II levels (local or plasma) would change in these conditions, if you could also discuss this fact.

Specific comments:

Title: please shorten and focus. Please omit drugs (captopril and L-NAME) and mention mechanism in title. It is also an issue that captopril is used in patient therapy but L-NAME is not. Also, nowhere in the text L-NAME is explained exactly: full name, enzyme of inhibition.

Abstract: is extremely long. Shorten itt o 200 words and please focus. Some uncertain phrases need to be rewritten (please allow me not to calculate the line numbers on the pages!):

„works asymmetrically,

modified under such conditions,

hypothalamus (HT), pituitary (PT), adrenal (AD) axis”: if „axis” is mentioned, a common shortening is better: HPA

„a hypertensive agent such as L-NAME” :  you have to specify the exact name and effect of it!

„between left FC and tissues of axis

For intra-tissue correlations

 In general, there is an increase in all of them”, etc.

Page 2, Introduction: please correct and specify:

Our organism

L-NAME

chapter 1.1: The whole chapter needs a rephrase and shortening according to major comment No. 2. Please omit Figures from here. Although, Fig 2 is suggested to put into the Discussion by modifications according to the observed changes in different areas.

1.2. Objectives

Please indicate here the aims and backgrounds of the present study clearly.

please rephrase:

„Partial data have previously been described ,

We currently have enough data that allow us to assess,

hypertensive treatment with the administration of L-NAME,

which can offer us a global idea about the tendency in the interactions between these tissues with an opposite hypotensive treatment against another hypertensive one, in genetically hypertensive animals”

angiotensin-converting enzyme is suggested to abbreviate also as ACE

Page 4, Results:

Pls explain „sol” and „m-b”

Pls specify: The rest were positive

(tables 1A and 1B) please correct here and later to Supplementary Tables.

Figure 3 Legend: Please indicate the measured parameters which were correlated. Please correct the references to Supplementary Tables.

Please rephrase: If we analyze all the correlation results

Page 7, Figure 4 legend: Please indicate the measured parameters which were correlated. Please correct the references to Supplementary Tables. Please give explanation what percentages are plotted in this figure, what parameter gives the 100 % in the panels.

Discussion:

„The animals used in the present investigation showed the following blood pressure levels with the various treatments: Control SHR: 206.7 ± 6.8 mmHg, Captopril: 159.3 ± 5.2 mmHg, L-NAME: 234.7 ± 4.1 [14].”  NEEDS TO BE PUT IN THE RESULTS. Also please indicate if these data were systolic or mean BP values. Please discuss why these data are so high and if animals had any other symptoms from these high blood pressure values.

„Control: • Significant negative correlations between right FC and the rest of the tis[1]sues. • Significant positive correlations between left FC and the rest of the tissues.

• Majority of significant positive inter-tissue correlations between HT, PT and AD. • The majority of significant intra-tissue correlations are positive. Captopril: • In general, significant correlations are reduced. • Significant negative correlations with right FC and positive with left FC are maintained. • Some slight significant inter-hemispheric correlations appear. • There is a notorious decrease in the significant inter-tissue correlations between HT, PT and AD. • The majority of significant intra-tissue correlations are positive. L-NAME: • A generalized disruption of significant correlations is observed in relation to the other two groups. • There is a general increase in significant inter-tissue and intra-tissue corre[1]lations. • The predominance of significant negative correlations between FC and the rest of tissues changes to the left FC in comparison to the other two groups. • All the significant intra-tissue correlations are positive, especially in right FC and HT where the number and level of correlations increase.” PLEASE WRITE COMPLETE SENTENCES OR GIVE THESE SUMMARY DATA IN A SUMMARY TABLE.

Figure 5 legend : please refer to Suppl. Tables.

Page 8. „Some of these partial results between PT and AD have already been observed previously [16].”  Please rephrase these sentences according to suggestions in major concern No 5.

Page 9 Conclusions line 1: Please rephrase „improvement ”

Page 9 Methods:

„Control SHR, captopril treated SHR and L-NAME treated SHR. Please give animal numbers of each group.

„Doses and timing of administration have been previously described [23, 24].

It is also suggested to indicate more methdodological details here.

„systolic blood pressure was determined by tail-cuff plethysmography in unanaesthetised rats. Please show these values in Results.

Specific soluble and membrane-bound aminopeptidase activities were ex[1]pressed as a nmol of the corresponding substrate hydrolyzed per minute per milligram of protein.”

One cannot see these data in the paper.

„Table 1A: Positive or negative correlations between the angiotensinase activities analyzed in the right or left frontal cortices versus angiotensinase activities analyzed in hypothalamus, pituitary or adrenal gland in control SHR. Table 1B: Positive or negative correlations between the angiotensi[1]nase activities analyzed in hypothalamus, pituitary or adrenal gland versus angiotensinase activi[1]ties analyzed in hypothalamus, pituitary or adrenal gland in control SHR. Table 2A: Positive or negative correlations between the angiotensinase activities analyzed in the right or left frontal cor[1]tices versus angiotensinase activities analyzed in hypothalamus, pituitary or adrenal gland in captopril treated SHR. Table 2B: Positive or negative correlations between the angiotensinase ac[1]tivities analyzed in hypothalamus, pituitary or adrenal gland versus angiotensinase activities an[1]alyzed in hypothalamus, pituitary or adrenal gland in captopril treated SHR. Table 3A: Positive or negative correlations between the angiotensinase activities analyzed in the right or left frontal cor[1]tices versus angiotensinase activities analyzed in hypothalamus, pituitary or adrenal gland in l-name treated SHR. Table 3B: Positive or negative correlations between the angiotensinase activi[1]ties analyzed in hypothalamus, pituitary or adrenal gland versus angiotensinase activities analyzed in hypothalamus, pituitary or adrenal gland in l-name treated SHR”

Please put this into the supplement.

Author Response

Dear reviewer:

Thank you for your comments.

Dear Editors

The paper needs an extensive revision throughout the text. Though the analysis is extensive, new data calculations and presentations are also suggested. An extensive English spell check revision is required.

I have tried to address all issues in my comments to consider.

Comments on the Quality of English Language

Review

Authors in their manuscript investigate angiotensinase activities in brain cortex and HPA tissues in SHR rats treated with captopril and L-NAME. They found significant changes in correlation between brain FC sides and HPA tissues as indicated of Fig. 4. They have made an extensive correlation analysis form their data.

They conclude that the benefits of treatment of hypertension with captopril may therefore convey a modification in the bilateral neu[1]ro-endocrine behavior of neuro-visceral integration.

Since the study is interesting, the manuscript is not well written, not focused, the presentation of data is not clear and not enough in the present form. The sections have to be focused and cleared. The paper is full of uncertain expressions, e.g. „modified under such conditions”, „In general, there is an increase in all of them”, „We currently have enough data that allow us to assess”, „The rest were positive”, etc, which are better for a personal blog, than for a scientific article.

Modified in the revised manuscript.

There are some major issues of the manuscript:

1.The manuscript does not apply for some formal requirements of IJMS: The abstract is too lengthy (200 words are allowed), it is almost double of that. The Conclusions is after Methods. The template does not contain line numbers on the pages, so it is more difficult for the reviever to interpret suggested corrections. The Supplementary material (Table legends) should appear in the Supplement, not in the main text. Etc.

The abstract has been shortened to 198 words. The other comments have also been contemplated in the revised Ms.

2.The Introduction looks like a book chapter for university students. It is not focused at all. Please indicate here the up-to-date scientific information about renin-angiotensin system and peptidases studied and their roles also with recent publications. Please omit figures from here and those detailed descriptions of the systems with more well known information.

The introduction has been reduced and Figure 1 eliminated in the revised Ms.

  1. A major issue of the study that what do readers see from data of pure correlation presentation, since there are original data presented. From pure correlations we do not know the sites of changes. It is suggested to show also original data from each tissue and show the main significant data as correlation curves of orig. data (as it is presented from previous publication of Authors in ref. 16).

The present paper is a correlational analysis of aminopeptidase activities involved in the Renin-Angiotensin System (RAS) in spontaneously hypertensive rats (SHR) treated with captopril and L-NAME in left Frontal Cortex, right Frontal Cortex, hypothalamus, pituitary and adrenal

  1. It is also the question why Authors show an investigation only among hypertensive animals. It would be useful to add a normotensive group for accurate comparisons. It is also required to show blood pressure data of the animal groups in Results and to discuss these data compared to previous studies. Please explain if systolic or mean values are indicated in the test and why these values are measured so high (above 200 mmHg, a data distribution in all groups would be reguired).

Unfortunately, data of the left and right frontal cortices of normotensive animals belonging to the original project were accidentally lost. The values in the left and right frontal cortex were reanalyzed in new normotensive animals, but the rest of the tissues were not analyzed. Therefore, these values could no longer be correlated with the HT, PT and AD values of normotensives and the complete and adequate correlational analysis can only be done with hypertensive animals.

  1. As it is mentioned on page 8 „Some of these partial results between PT and AD have already been observed previously [16].”, it needs some clarification if Authors used data from previous publications or if they have not published the present data before. In a prev. case maybe it needs some permission requirements.

The present study: analysis correlational between LFC, RFC, HT, PT, AD is original.. The rest is informative and appropriately referenced.

  1. The references are full of older papers. Please give also more recent publications of the area and add to them.

Some new references have been added in the revised Ms.

  1. It is also a question how angiotensin II levels (local or plasma) would change in these conditions, if you could also discuss this fact.

The regulation of Ang II may be performed in part by the influence of captopril and L-NAME treatments on GluAP.

Specific comments:

Title: please shorten and focus. Please omit drugs (captopril and L-NAME) and mention mechanism in title. It is also an issue that captopril is used in patient therapy but L-NAME is not. Also, nowhere in the text L-NAME is explained exactly: full name, enzyme of inhibition.

Captopril and L-NAME were used as hypotensive or hypertensive agents. This has been modified in the revised Ms.

Abstract: is extremely long. Shorten itt o 200 words and please focus. Some uncertain phrases need to be rewritten (please allow me not to calculate the line numbers on the pages!):

„works asymmetrically,

modified under such conditions,

hypothalamus (HT), pituitary (PT), adrenal (AD) axis”: if „axis” is mentioned, a common shortening is better: HPA

„a hypertensive agent such as L-NAME” : you have to specify the exact name and effect of it!

„between left FC and tissues of axis

For intra-tissue correlations

In general, there is an increase in all of them”, etc.

The abstract has been modified and shortened now to 198 words.

Page 2, Introduction: please correct and specify:

Our organism

L-NAME

chapter 1.1: The whole chapter needs a rephrase and shortening according to major comment No. 2. Please omit Figures from here. Although, Fig 2 is suggested to put into the Discussion by modifications according to the observed changes in different areas.

Modified in the revised Ms.

1.2. Objectives

Please indicate here the aims and backgrounds of the present study clearly.

please rephrase:

„Partial data have previously been described ,

We currently have enough data that allow us to assess,

hypertensive treatment with the administration of L-NAME,

which can offer us a global idea about the tendency in the interactions between these tissues with an opposite hypotensive treatment against another hypertensive one, in genetically hypertensive animals”

angiotensin-converting enzyme is suggested to abbreviate also as ACE

Modified in the revised Ms.

Page 4, Results:

Pls explain „sol” and „m-b”

Pls specify: The rest were positive

(tables 1A and 1B) please correct here and later to Supplementary Tables.

Modified in the revised Ms.

Figure 3 Legend: Please indicate the measured parameters which were correlated. Please correct the references to Supplementary Tables.

Figure 3 has been eliminated in the modified Ms. Figures 3 and 4 have been removed. Instead, we have included three figures showing the significant correlations between the left or right frontal cortex with the hypothalamus, pituitary and adrenal cortex in the three groups studied.

Please rephrase: If we analyze all the correlation results

Eliminated in the revised Ms.

Page 7, Figure 4 legend: Please indicate the measured parameters which were correlated. Please correct the references to Supplementary Tables. Please give explanation what percentages are plotted in this figure, what parameter gives the 100 % in the panels.

Figure 4 has been eliminated in the modified Ms.

Discussion:

„The animals used in the present investigation showed the following blood pressure levels with the various treatments: Control SHR: 206.7 ± 6.8 mmHg, Captopril: 159.3 ± 5.2 mmHg, L-NAME: 234.7 ± 4.1 [14].”  NEEDS TO BE PUT IN THE RESULTS. Also please indicate if these data were systolic or mean BP values. Please discuss why these data are so high and if animals had any other symptoms from these high blood pressure values.

Modified in the revised Ms.

„Control: • Significant negative correlations between right FC and the rest of the tis[1]sues. • Significant positive correlations between left FC and the rest of the tissues.

  • Majority of significant positive inter-tissue correlations between HT, PT and AD. • The majority of significant intra-tissue correlations are positive. Captopril: • In general, significant correlations are reduced. • Significant negative correlations with right FC and positive with left FC are maintained. • Some slight significant inter-hemispheric correlations appear. • There is a notorious decrease in the significant inter-tissue correlations between HT, PT and AD. • The majority of significant intra-tissue correlations are positive. L-NAME: • A generalized disruption of significant correlations is observed in relation to the other two groups. • There is a general increase in significant inter-tissue and intra-tissue corre[1]lations. • The predominance of significant negative correlations between FC and the rest of tissues changes to the left FC in comparison to the other two groups. • All the significant intra-tissue correlations are positive, especially in right FC and HT where the number and level of correlations increase.” PLEASE WRITE COMPLETE SENTENCES OR GIVE THESE SUMMARY DATA IN A SUMMARY TABLE.

Modified in the revised Ms.

Figure 5 legend : please refer to Suppl. Tables.

Included

Page 8. „Some of these partial results between PT and AD have already been observed previously [16].”  Please rephrase these sentences according to suggestions in major concern No 5.

See Major concern No 5.

Page 9 Conclusions line 1: Please rephrase „improvement ”

Modified in the revised Ms.

Page 9 Methods:

„Control SHR, captopril treated SHR and L-NAME treated SHR. Please give animal numbers of each group.

This was already indicated in 4.1. of Materials and methods.

„Doses and timing of administration have been previously described [23, 24].

It is also suggested to indicate more methdodological details here.

This was already indicated in 4.1. of Materials and methods.

„systolic blood pressure was determined by tail-cuff plethysmography in unanaesthetised rats. Please show these values in Results.

This is now indicated in Results.

Specific soluble and membrane-bound aminopeptidase activities were ex[1]pressed as a nmol of the corresponding substrate hydrolyzed per minute per milligram of protein.”

One cannot see these data in the paper.

The objective of the present paper are correlational values. Such data were previously reported [14-16]. There was a mistake in this sentence (the values were not nmol but pmol) that has been now modified.

„Table 1A: Positive or negative correlations between the angiotensinase activities analyzed in the right or left frontal cortices versus angiotensinase activities analyzed in hypothalamus, pituitary or adrenal gland in control SHR. Table 1B: Positive or negative correlations between the angiotensi[1]nase activities analyzed in hypothalamus, pituitary or adrenal gland versus angiotensinase activi[1]ties analyzed in hypothalamus, pituitary or adrenal gland in control SHR. Table 2A: Positive or negative correlations between the angiotensinase activities analyzed in the right or left frontal cor[1]tices versus angiotensinase activities analyzed in hypothalamus, pituitary or adrenal gland in captopril treated SHR. Table 2B: Positive or negative correlations between the angiotensinase ac[1]tivities analyzed in hypothalamus, pituitary or adrenal gland versus angiotensinase activities an[1]alyzed in hypothalamus, pituitary or adrenal gland in captopril treated SHR. Table 3A: Positive or negative correlations between the angiotensinase activities analyzed in the right or left frontal cor[1]tices versus angiotensinase activities analyzed in hypothalamus, pituitary or adrenal gland in l-name treated SHR. Table 3B: Positive or negative correlations between the angiotensinase activi[1]ties analyzed in hypothalamus, pituitary or adrenal gland versus angiotensinase activities analyzed in hypothalamus, pituitary or adrenal gland in l-name treated SHR”

Please put this into the supplement.

Now in the supplementary material.

Reviewer 2 Report

Comments and Suggestions for Authors

In the submitted paper the authors studied the enzymatic activities of three different aminopeptidases in tissue homogenates from the left and right frontal cortex, hypothalamus, pituitary and adrenal glands   of spontaneously hypertensive rats (SHR) and computed correlations between activities found in different areas as well as between both sides of the brain. Membrane bound (supernatant) and cytoplasmic activities were separately measured. Measurements were made under conditions of chronically elevated and decreased blood pressure to demonstrate blood pressure effects on brain asymmetry and on correlation between one of the hemispheres with members of the hypothalamo-pituitary-adrenal axis. The work is a continuation of a series of earlier publications from the same group in which the asymmetry of angiotensinase activity in different areas was demonstrated in normal rat brain (Banegas I et al Behav Brain Res 2005;156:321) and in normal and spontaneously hypertensive animals whom 6-hydroxydopamine injection was given at one side into the striatum (Banegas I et al. Pharmacol Biochem Behav 2019;182:12).

Demonstration of aminopeptidase activity in the frontal cortex of the  brain and in the hypothalamo-pituitary-adrenal axis  in hypertensive animals, its alteration with blood pressure are important achievements, however, the logical buildup of the paper and conclusions drawn can be criticized.

The three aminopeptidases, the activity of which was studied with a specific chromophore might not be specific for angiotensin. Their presence misleadingly is connected to the presence of angiotensin, the expression of which was not studied. One would suggest even to change the title of the paper, for  “aminopeptidases”  while “angiotensinase” put in brackets. Later, it should be analyzed what substances can serve as substrates at different locations. A limitation sentence should be added to the paper.

Also it is not fully clear, that if to demonstrate biochemical asymmetry of the rodent brain in hypertension was the aim, why this particular transmitter and why the enzymes in its degradation have been chosen? This does not affect the value of their basic observations, that there exist several asymmetries in aminopeptidase activities in cortical structures in spontaneously hypertensive rats which show alteration if blood pressure is changing and which in several ways are correlating in an asymmetric manner with hypothalamic, pituitary and adrenal activities of the same enzymes. Some effort, however, should be exerted to explain, why aminopeptidase activities can characterize the activity of the RAS?

Not fully understandable is the logic of the in vivo application of LNAME and captopril. What a mechanism of action the authors did expect? Blood pressure changes should have been induced by some other method with less potential interference with the RAS (captopril) and without a direct effect on the vessels of the brain (LNAME).  Chronic  pressure changes are expected to effect the expression of aminopeptidases? If it is by some posttranslational modification, what exactly it is? If composition of the cytoplasm is responsible for the enzyme activity alteration– such will massively change during the biochemical preparation process. Limitation remarks should be added to make the readers intent on this potential problems.

We are offered an impressive array of positive and negative correlations identified between aminopeptidase activities in left and right cortex, hypothalamus, pituitary and adrenal glands. An important, stable achievement is that even in the rodent, there exists a left-right asymmetry of the cortex in this respect. One serious shortfall of the paper, however,  is that we do not get any potential explanation for such interactions between so distantly located areas of the body. Do the authors hint at the potential existence of an obscure, unidentified yet contact between the RAS components in different areas of the brain and body? Being even under cortical control? Or some hormone or neural pathway could synchronize that element of the local angiotensin degradation? One more simple explanation is that blood pressure itself synchronizes RAS components in different parts of the body, and of that altered RAS activity is one, if not the most important component the observed angiotensinase activity.  Studying alterations in more components of the RAS, or even studying the plasma levels of more hormones, recording neural activities toward or inside the hypothalamus could have helped remove that mist. A limitation statement should be added, according to further studies are needed to reveal the known or unknown pathways resulting the observed (asymmetric) correlations.

 Early quantitative histology observations failed to find any difference between the density of myelinated and non-myelinated fibers in the white matter of the rat forebrain (Partadiredja G et al. J Neurocytol  2003;32:1165). Cite as potential limitation.

A paper published by Wu HM (Neuropeptides 2010;44:253) found a correlation between  the paw preference and Cys-aminopeptidase and Ala-aminopeptidase activity in the hippocampus of normal rats, it should be included in the evaluation.

Left-right asymmetry has been proven in proteomic studies by Samara (Hyppocampus 2011;21:108) for the expression of more than 50 proteins of the rat hippocampus. Can the authors see any connection with their findings?

Alanine aminopeptidase has been found to be released from astrocytes and stimulate chronic inflammatory reactions in microglia (Kim JH et al. Mol Cell Proteomics 2022;21:100424). Any connection with the presented observations?

Some specific remarks

Introduction. Should be massively shortened. The paper does not substantiate such an elaborate and didactic description of proteases. Instead, something about the specificity of the aminopeptidases under study should be written. More attention should be paid to potential control mechanisms of aminopeptidase activity (protein expression, activation processes, potential substrate concentrations). Some explanation should be offered why using the activity of these enzymes the authors attempted to describe RAS activity.

Results. Should be massively shortened. Listing correlation factors by numbers in the text, demonstrating them in Figures is too much. Pick up a few characteristic interconnections and demonstrate it on an easily perceivable Figure.

Discussion. Should be massively shortened. Keep Figure 5 and explain the most important interconnections observed.

Language, style, Figures of the paper reflect high levels of professionalism.

Author Response

Dear reviewer:

Thank you very much for your comments.

Addressing your comments, the general paper has been now extensively modified. In addition, figures 3 and 4 have been removed. Instead, we have included three figures showing the significant correlations between the left or right frontal cortex with the hypothalamus, pituitary and adrenal in the three groups studied. In addition, we have now discussed the references that you comment at the end of Discussion.

Round 2

Reviewer 1 Report

Comments and Suggestions for Authors

Please provide a text with showing all of the track changes (also all deletions and modifications) since the first submitted version! Authors can do it with a simple word comparison. Since the changes of the manuscript is impossible to follow this way as track changes do not show all changes and deletions.

Also please provide a final text version in the upload without any red changes. Showing citations in both versions is misleading (in lines 47,69, 74, 75, 77).

There are lot of improvements in the interpretation of data (Results) with correlation figures, but still an issue has been remained:

Related to this answer:

  1. It is also the question why Authors show an investigation only among hypertensive animals. It would be useful to add a normotensive group for accurate comparisons. It is also required to show blood pressure data of the animal groups in Results and to discuss these data compared to previous studies. Please explain if systolic or mean values are indicated in the test and why these values are measured so high (above 200 mmHg, a data distribution in all groups would be reguired).

"Unfortunately, data of the left and right frontal cortices of normotensive animals belonging to the original project were accidentally lost. The values in the left and right frontal cortex were reanalyzed in new normotensive animals, but the rest of the tissues were not analyzed. Therefore, these values could no longer be correlated with the HT, PT and AD values of normotensives and the complete and adequate correlational analysis can only be done with hypertensive animals."

Please include also normotensive data which you have on frontal cx as mentioned and also normotensive blood pressure data in the Results. Please extend Results with this if possible.

Minor comments: Please explain all abbreviations in each Figure legend, e.g. HT, PT, AD, SHR, NAME

Lines 174-198 Discussion: Please omit line markers in the text and please use continous sentences in the all of this chapter:

For example: 

In the control (SHR) group we have found that......

Also as "control" is a hypertensive control, you could also refer to normotensives here in some respect.

"Control: Only significant negative correlations between right FC and the rest of the tissues  were observed. • Only significant positive correlations between left FC and the rest of the tissues were observed. • A majority of significant positive inter-tissue correlations were observed between  HT, PT and AD.  • The majority of significant intra-tissue correlations were positive.  Captopril: • In general, the observed number of significant correlations were reduced. • A majority of significant negative inter-tissue correlations with right FC and  positive with left FC were maintained.  • Some slight significant inter-hemispheric correlations appeared.  • There wass a notorious decrease in the number of significant inter-tissue corre-lations between HT, PT and AD.  • The majority of significant intra-tissue correlations were positive.  L-NAME: • A generalized disruption of the positive or negative character and the number of significant correlations was observed in comparison with the other two groups. • There was a general increase in the number of significant inter-tissue and in- tra-tissue correlations.  • The highest number of significant negative correlations between FC and the rest  of tissues changed to be in the left FC in comparison with the other two groups.  • All the significant intra-tissue correlations were positive, especially in right FC  and HT where the number and level of correlations increased. "

Comments on the Quality of English Language

Minor check, some corrections needed.

Author Response

Dear reviewer,

Thank you for your comments.

Comments and Suggestions for Authors

Please provide a text with showing all of the track changes (also all deletions and modifications) since the first submitted version! Authors can do it with a simple word comparison. Since the changes of the manuscript is impossible to follow this way as track changes do not show all changes and deletions.

Enclosed is the "word comparison" between the original submission and the present one.

Also please provide a final text version in the upload without any red changes. Showing citations in both versions is misleading (in lines 47,69, 74, 75, 77).

Now corrected

There are lot of improvements in the interpretation of data (Results) with correlation figures, but still an issue has been remained:

Related to this answer:

  1. It is also the question why Authors show an investigation only among hypertensive animals. It would be useful to add a normotensive group for accurate comparisons. It is also required to show blood pressure data of the animal groups in Results and to discuss these data compared to previous studies. Please explain if systolic or mean values are indicated in the test and why these values are measured so high (above 200 mmHg, a data distribution in all groups would be reguired).

"Unfortunately, data of the left and right frontal cortices of normotensive animals belonging to the original project were accidentally lost. The values in the left and right frontal cortex were reanalyzed in new normotensive animals, but the rest of the tissues were not analyzed. Therefore, these values could no longer be correlated with the HT, PT and AD values of normotensives and the complete and adequate correlational analysis can only be done with hypertensive animals."

Please include also normotensive data which you have on frontal cx as mentioned and also normotensive blood pressure data in the Results. Please extend Results with this if possible.

Because of our study is addressed to a correlational analysis between left and right frontal cortex with the HPA axis in SHR, we believe that the inclusion of the data of frontal cortices and systolic blood pressure in normotensive animals may introduce confusion to the reader. We additionally have added an informative comment in the discussion (see page 7 lines 103-105). In any case, these data are already published [14] and appropriately referred several times in the Ms. In addition, the indicated blood pressures are systolic blood pressure values. See Materials and Methods (page 9 lines 176-179). Additionally, in the revised Ms. we have now reasoned the potential mechanisms by which L-NAME increased blood pressure (see page 3 lines 96-97, ref [17]).

Minor comments: Please explain all abbreviations in each Figure legend, e.g. HT, PT, AD, SHR, NAME.

Now explained

Lines 174-198 Discussion: Please omit line markers in the text and please use continous sentences in the all of this chapter:

Now modified as indicated.

For example: 

In the control (SHR) group we have found that......

Also as "control" is a hypertensive control, you could also refer to normotensives here in some respect.

"Control: Only significant negative correlations between right FC and the rest of the tissues  were observed. • Only significant positive correlations between left FC and the rest of the tissues were observed. • A majority of significant positive inter-tissue correlations were observed between  HT, PT and AD.  • The majority of significant intra-tissue correlations were positive.  Captopril: • In general, the observed number of significant correlations were reduced. • A majority of significant negative inter-tissue correlations with right FC and  positive with left FC were maintained.  • Some slight significant inter-hemispheric correlations appeared.  • There wass a notorious decrease in the number of significant inter-tissue corre-lations between HT, PT and AD.  • The majority of significant intra-tissue correlations were positive.  L-NAME: • A generalized disruption of the positive or negative character and the number of significant correlations was observed in comparison with the other two groups. • There was a general increase in the number of significant inter-tissue and in- tra-tissue correlations.  • The highest number of significant negative correlations between FC and the rest  of tissues changed to be in the left FC in comparison with the other two groups.  • All the significant intra-tissue correlations were positive, especially in right FC  and HT where the number and level of correlations increased. "

Comments on the Quality of English Language

Minor check, some corrections needed.

Reviewed again.

Submission Date

07 September 2023

Date of this review

16 Oct 2023 01:17:16

Round 3

Reviewer 1 Report

Comments and Suggestions for Authors

Only few minor remarks:

Please refer to Supplementary tables in lines 111,133,158.

Please explain abbrev. NAME in Fig. legends 4.

Please check references in the text.

Comments on the Quality of English Language

Only minor editing is required.

Author Response

Dear reviewer,

Thank you for your comments.

Only few minor remarks:

Please refer to Supplementary tables in lines 111,133,158.

Referred in the revised Ms.

Please explain abbrev. NAME in Fig. legends 4.

Specified now in Fig 5

Please check references in the text.

References checked in the text

Comments on the Quality of English Language

Only minor editing is required.

Checked again